# Coupling Characteristics of Powder and Laser of Coaxial Cone Nozzle for Laser Direct Metal Deposition: Numerical Simulation and Experimental Study

**DOI:** 10.3390/ma16093403

**Published:** 2023-04-26

**Authors:** Zhenhao Wang, Kaihua Hu, Lin Yang, Jian Zhang, Honghui Ding, Zelong Pan

**Affiliations:** 1College of Mechanical & Electrical Engineering, Wenzhou University, Wenzhou 325035, China; 2Pingyang Institute of Intelligent Manufacturing, Wenzhou University, Wenzhou 325409, China; 3Zhejiang University—University of Illinois at Urbana-Champaign Institute, Zhejiang University, Haining 325006, China; 4School of Mechanical & Automotive Engineering, Qingdao University of Technology, Qingdao 266520, China

**Keywords:** laser direct metal deposition, numerical simulation, coaxial cone nozzle, gas–solid flow, coupling characteristics

## Abstract

Laser direct metal deposition (LDMD) enables not only the preparation of high-performance coatings on the surfaces of low-property materials but also the three-dimensional direct manufacturing and re-manufacturing of parts. In the LDMD process, the spatial coupling characteristics of the powder flow and the laser beam are the key factors affecting the forming quality of the cladding layer. Based on the gas–solid two-phase flow theory, a numerical model of coaxial powder feeding was established by CFD. The powder flow characteristics of the lower part of the nozzle, the powder particle motion trajectory, and the optical-powder spatial coupling morphology and law were studied, and the relationship between the powder flow morphology, laser beam, and powder utilization was explored. On this basis, the law between the optical-powder coupling characteristics and the geometric characteristics of the cladding layer is discussed in conjunction with LDMD experiments. The results show that the powder concentration scalar located in the focal plane of the laser beam can be used to visualize the optical-powder coupling morphology. When the powder feeding speed exceeds the loading capacity of the carrier gas flow, the powder concentration in the center of the spot and the powder utilization rate decrease. When the carrier gas flow rate is 4.0 L/min and the powder feeding rate is 4.0 g/min, the best utilization rate achieved is 81.4%. In addition, the *H* (height) of the cladding layer is more sensitive to changes in the powder concentration than the *W* (width). These findings provide new ideas for nozzle structure design and the optimization of LDMD parameters.

## 1. Introduction

Laser direct metal deposition (LDMD) is a process of transporting metal powder to the substrate surface through one or several nozzles. Under the action of a laser, the metal powder is melted to the substrate surface in a layer-by-layer manner and undergoes metallurgical bonding with the substrate [1,2,3]. Multiple layers of molten metal powder are deposited and stacked together, and this process can be used to prepare high-performance isotropic coatings on the surfaces of related components, quickly repair damaged components, and even realize the manufacturing of complex three-dimensional geometric components [4,5,6]. In addition, the coaxial nozzle is the key powder feeding component in the LDMD process. The trajectory and concentration distribution of the powder particles at the bottom of the nozzle directly affect the powder utilization and deposition quality [7,8,9]. To accurately describe the coupling between the powder particles at the bottom of the nozzle and the laser, detailed analysis, calculation methods, and process diagnosis are required.

The structure of the nozzle affects the flow of the powder particles, which, in turn, affects the performance of the prepared coating [10,11,12]. However, the flow of the powder particles can be analyzed using fluid dynamics. Complex flow characteristics are ambiguous if one relies on experimental results alone [13]. Therefore, considering the complex multiphase flow in the coaxial nozzle, Zekovic et al. [14] proposed to accurately simulate it using numerical modeling techniques. On this basis, Wu et al. [15] built a powder flow distribution model with a four-jet coaxial nozzle and found that the injection radius and angle of the nozzle had significant impacts on the focal spot characteristics of the powder flow. In addition, the particle size, powder density, powder shape factor, and powder flow rate set using the model are all key factors that affect the mass concentration of the powder flow [16]. Mahmood et al. [17] developed a three-jet coaxial finite element model, with which they detected the gradual transition of the powder flow from a Gaussian to an annular distribution in line with increments in the distance from the focus point. Liu et al. [18] used numerical simulations to study the effect of the powder supply rate on the concentration of nickel alloy powder particles between the coaxial nozzle and the substrate and found that the powder concentration had a positive correlation with the powder supply rate. Guo et al. [19] established a coaxial nozzle model to study the influence of powder flow with different carrier gas flows on the external flow field of the nozzle. It was found that the larger the carrier gas flow was, the higher the flow field velocity at the nozzle outlet was. The above research has mainly been based on the coaxial nozzle structure, aiming to study which factors affect the powder flow characteristics and powder particle concentration. However, for the 3D manufacturing and re-manufacturing of parts, a more precise and finer coaxial conical nozzle structure is required.

Compared to coaxial nozzles with radial symmetry (three-jet [20] and four-jet nozzles [21]), the coaxial cone nozzle can provide an accurate powder supply speed in any direction, making it suitable for the 3D cladding of complex motion trajectories [22]. Ferreira et al. [23] studied the powder flow characteristics under different coaxial conical nozzle designs. The results showed that only the shaping gas, powder flow rates, and nozzle design impact the powder stream diameter. Pan et al. [24] investigated the influences of nozzle geometry, shielding gas, and powder characteristics on the particle concentration of the powder flow through the mathematical modeling of the coaxial cone nozzle. Momentum loss was caused by the interaction between the in-flight powder particles and the collision between the particles and the nozzle wall. The momentum loss could determine the shape of the powder flow, as well as the trajectory and mass concentration of the powder particles [25,26]. Kovalev et al. [27,28] introduced the momentum recovery coefficient into a numerical model, and it was found that the inelastic collision between the powder particles and the inner nozzle wall reduced the speed of the powder particles, which was conducive to the convergence of the powder flow. It is noteworthy that in the LDMD process, the effective utilization of powder particles is determined by the mass concentration of the powder particles in the coupling plane between the powder flow and the laser beam focal plane, but few studies have been conducted on the numerical simulation of this part. Therefore, it is highly important to further explore the coupling characteristics of the focal plane of the powder flow and laser beam, especially the mass concentration of the powder particles in the coupling plane, in order to explain the utilization of the powder.

In this work, the coaxial cone nozzle, which was employed in an experimental study, was selected as a prototype for numerical simulation. Based on the experimental verification of powder flow characteristics, the coupling characteristics of the 3D powder particle trajectory and the laser were studied, thus providing new ideas for the optimization of nozzle structures and process parameters.

## 2. Numerical Modeling

### 2.1. Physical Model Statement

A structural diagram of the coaxial cone nozzle is shown in Figure 1. The overall structure of the nozzle is composed of four-part concentric cones. The four-way carrier gas powder channel is located inside the upper cone (3). The lower cones (2 and 4) form a conical channel for the powder flow through the matching gap, and the vertical height difference between the two is 1.3 mm, which achieves the effect of the powder flow shape. At the same time, in order to avoid overheating of the nozzle, a circulating cooling channel is included inside the lower cones (2 and 4). The specific geometric parameters of the structure are shown in Table 1.

### 2.2. Calculation Parameters

ICEM CFD12.1 software was used to generate the mesh of the coaxial conical nozzle model shown in Figure 2. Due to the complex structure of the nozzle model, an unstructured mesh should be used: a hexahedral mesh for the cylindrical region, a tetrahedral mesh for the nozzle, and the junction between the nozzle and the cylinder. The domain calculation consists of two parts: the coaxial cone nozzle and the domain expansion. The domain expansion is defined as a cylinder of 62 mm in diameter and 56 mm in height, and its boundary is the escape wall. The initial conditions for the calculation are as follows: temperature T = T_0_ = 300 K, with an initial pressure and escape pressure of 100 kPa and 101.35 kPa, respectively.

The powder particle selected for the numerical simulation is Ni60A, and its detailed simulation parameters are shown in Table 2. In order to facilitate the numerical simulation, the average particle size of the powder particles was used as one of the simulation parameters for the discrete phase (Ni60A), while the average particle size of the powder was measured using a laser particle-size analyzer. The average value of the particle size was measured with a laser granularity analyzer, and this value was used to solve the problem. First, the dispersion medium and tested powder are placed into the instrument, and the ultrasonic generator is started up to fully disperse the samples. Then, the circulation pump is started, and one waits for the end of the test. Finally, the particle size data are output using the instrument’s supporting software. The momentum loss, which is evaluated using the restitution coefficient (*k_n_*, 0.9–0.99) [29], is caused by the collision interactions between the in-flight powder particles and the nozzle wall. In this study, the elastic collision and the inelastic collision were 1.0 and 0.9, respectively. Argon was used as the carrier gas, and its viscosity and density were 2.125 × 10^−5^ kg/ms and 1.6228 kg/m^3^, respectively. The velocity of the carrier gas at the nozzle inlet is defined in Equation (1) [30], where *v* is the velocity of the carrier gas, *Q* is the flow rate of the carrier gas, and *S* is the cross-sectional area of the nozzle inlet.
(1)$$v=\frac{Q}{4S}$$

### 2.3. Method of Equation Solution and General Assumptions

The powder particles used in this study were discrete solid particles and could be regarded as a discrete phase. They enter the nozzle in a four-way process under the thrust of the carrier gas. In this work, the carrier gas was set as a continuous phase and was solved using the Navier–Stokes equations, while the powder particles were regarded as a discrete phase and were solved using the Euler coordinate system with the Lagrangian equation. The equations that were used to solve the mass conservation and momentum conservation of the continuous phase are defined as [12,31,32]:(2)$$\frac{\partial \rho_{g}}{\partial_{t}}+\nabla \cdot (\rho_{g}v_{g})=0$$
(3)$$\frac{\partial \rho_{g}u}{\partial_{t}}+\nabla (\rho_{g}v_{g}u)=-\frac{\partial p_{g}}{\partial x}+\nabla (\mu \nabla u)+f_{x}$$
(4)$$\frac{\partial \rho_{g}v}{\partial_{t}}+\nabla (\rho_{g}v_{g}v)=-\frac{\partial \rho_{g}}{\partial y}+\nabla (\mu \nabla v)+f_{y}$$
(5)$$\frac{\partial \rho_{g}w}{\partial_{t}}+\nabla (\rho_{g}v_{g}w)=-\frac{\partial \rho_{g}}{\partial z}+\nabla (\mu \nabla w)+f_{z}$$
where *ρ_g_* is the density of the carrier gas; *v_g_* is the defined velocity of the gas field, which is composed of Cartesian velocities (*u*, *v*, *w*); *u*, *v*, and w represent the velocity vectors of the carrier gas in the *x*, *y*, and *z* directions, respectively; *P_g_* is the pressure of the carrier gas; *µ* is the viscosity of the carrier gas; and *f_x_*, *f_y_*, and *f_z_* are the sources of momentum in the *x*, *y*, and *z* directions, respectively, per unit volume and unit time.

Due to the fact that the flow of the carrier gas in the gas–solid flow is considered a turbulent state, the standard *k-ε* model equation is used to solve it [33]. The conservation equation of turbulence (*k* equation) is defined as follows:(6)$$\frac{\partial }{\partial x_{i}}(\rho k{\bar{\mu }}_{i})=\frac{\partial }{\partial x_{i}}(\frac{\mu_{t}}{\sigma_{k}}\frac{\partial k}{\partial x_{i}})+G_{k}+G_{b}-\rho \varepsilon$$

The conservation equation of dissipation of turbulence (*ε* equation) is defined as follows:(7)$$\frac{\partial }{\partial x_{i}}(\rho \varepsilon {\bar{\mu }}_{i})=\frac{\partial y}{\partial x_{i}}(\frac{\mu_{t}}{\sigma_{\varepsilon }}\frac{\partial \varepsilon }{\partial x_{i}})+C_{1\varepsilon }\frac{\varepsilon }{k}(G_{k}+G_{b})-C_{2\varepsilon }\rho \frac{\varepsilon^{2}}{k}$$
(8)$$G_{k}=\mu_{t}(\frac{\partial {\bar{u}}_{j}}{\partial x_{i}}+\frac{\partial {\bar{u}}_{i}}{\partial x_{j}})\frac{\partial {\bar{u}}_{j}}{\partial x_{j}}$$
(9)$$G_{b}=-g_{i}\frac{\mu_{t}}{\rho {\mathrm{Pr}}_{t}}\frac{\partial \rho }{\partial x_{i}}$$
where *k* is the kinetic energy of turbulence; *ε* is the dissipation rate of the kinetic energy; *G_k_* is the generation rate of the kinetic energy, which depends on the average velocity gradient; *G_b_* is the kinetic energy of turbulence due to buoyancy; and *Pr_t_* is the Prandtl number of turbulences, defined as Equation (10), where *σ_k_* = 1.0, *σ_ε_* = 1.3, *C*_1*ε*_ = 1.44, *C*_2*ε*_ = 1.92, and *C_u_* = 0.09 are empirical constants [34].
(10)$$\mu_{t}=\rho C_{u}\frac{k^{2}}{\varepsilon }$$

The powder particles are driven by the drag of the gas flow. In the Lagrangian reference frame, the trajectory of particle movement can be obtained by integrating both the inertia of each particle and the drag acting on the particle. The integrating equation of the force balance is defined as follows [23]:(11)$$\frac{dv_{p}}{dt}=\frac{m_{p}}{\tau_{p}}(v-v_{p})$$
(12)$$\tau_{p}=\frac{4\rho_{p}d_{p}{}^{2}}{3\mu C_{D}R_{e}}$$
(13)CD=a1+a2Re+a3Re2
(14)$$R_{e}=\frac{\rho_{p}D_{p}|v_{p}-v_{g}|}{\mu }$$
where *v_p_* is the velocity of the particles; *m_p_* is the quality of the particles; *a*_1_, *a*_2_, and *a*_3_ are empirical constants; *ρ_p_* is the density of the particles; *d_p_* is the diameter of the particles; and *R_e_* is the Reynolds number. Equations (2)–(14) were solved using the Pressure-Based solver and the SIMPLE algorithm.

The concrete assumptions for the calculation model are as follows:

The carrier gas and the powder particles are evenly distributed before entering the nozzle, and the velocity vectors of both media are the same. The powder particles are homogeneous and spherical, and their size is regarded to conform to the Rosin–Rammler distribution expression [26,35].

The carrier gas was set as a compressible, viscous, and steady-state continuum turbulence.

Only the forces of drag, inertia, and gravity were considered, while the other forces were ignored. We also ignored the feedback effect of the powder particles on the gas in motion and the thermal effect of laser radiation on the powder.

## 3. Experimental Method

The laser cladding tests were performed using a Rofin-FL060 6 kW continuous fiber laser processing system, as shown in Figure 3a. The laser cladding head (HIGHYAG BIMO, Germany) and the coaxial cone nozzle (Figure 3c) were mechanically matched and fixed onto the robot arm (ABB IRB4600, Switzerland). The laser light emitted by the laser was transmitted to the collimating and focusing system of the cladding head through the optical fiber, and the flat-top beam of 6 × 6 mm^2^ was output. We used a negative ballast air powder feeder (GTV PF2/2, Germany) to convey the powder particles and adjusted the feeding amount of the powder by changing the speed of the powder pan. The selected alloy powder was Ni60A, the particle diameter ranged from 35 to 100 µm (the average size was 79 µm), and the microscopic morphology is shown in Figure 3b. The substrate selected for the laser cladding test was AISI 1045 steel (60 × 20 × 8 mm^3^). The geometric characteristics of the cladding layer obtained by single laser cladding and the actual utilization of powder particles were compared with the light of the coaxial cone nozzle. The simulation results of the powder coupling were verified. The specific process parameters of the test are shown in Table 3.

An M110 high-speed camera was used to capture the powder flow morphology during the laser cladding process. The actual utilization of the powder particles was calculated according to the weighing method. After the laser cladding test, wire cutting was used to cut the cladding coating of the substrate along the cross-section in order to prepare a metallographic sample. We placed the prepared metallographic sample in an ultrasonic machine equipped with acetone solution for cleaning and degreasing, and then we used SiC sandpaper to grind and polish the surface (the polishing agent particle size was 1 μm). We cleaned the polished surface with absolute ethanol and then used a 4% nitric acid–alcohol mixed solution to corrode the polished surface. Finally, an optical microscope (OLYMPUS BX53M, Tokyo, Japan) was used to observe the geometric characteristics of the cross-section.

## 4. Results and Discussion

### 4.1. Experimental Verification of Powder Flow Characteristics

Figure 4 shows the morphology and concentration of the powder flow at the lower part of the coaxial cone nozzle under different recovery coefficients (*k_n_* = 0.9 and 1.0, carrier gas flow: 4.0 L/min, powder feeding rate: 4.0 g/min). When *k_n_* = 0.9, the powder particle concentration in the center of the focal plane is 2.5 kg/m^3^, which is significantly higher than the value of 2.26 kg/m^3^ obtained when *k_n_* = 1.0. Due to the momentum loss caused by inelastic collisions, the velocity of the powder particles at the nozzle outlet decreased, and their flight trajectory was closer to the carrier gas trajectory, thereby achieving better convergence effects. This experimental phenomenon is consistent with the research results of Liu et al. [30]. The predicted powder flow morphology during inelastic collisions (Figure 4a) is in close agreement with the experimental results (Figure 4c), which further proves that the model can be used to study the light-powder coupling characteristics of coaxial cone nozzles.

### 4.2. Coupling Characteristics of the 3D Powder Particle Trajectory and Laser

Figure 5 shows the coupling results of the axial powder particles’ trajectory and laser beam under different carrier gas flow and powder feeding rates obtained through numerical calculations. The coupling shape of the powder particles and the laser beam in space was affected by both the carrier gas flow and powder feed rates. The powder particles were ejected from the cone channel under the drag effects of the carrier gas and gravity. The focal plane of the obtained powder flow was located between 7.9 mm and 15.0 mm below the nozzle, which was consistent with the axial distance from the two peaks of the concentration of the powder flow to the nozzle (Figure 4d). The focal plane of the laser beam was settled 8 mm below the nozzle to reduce the shielding of the laser beam by the powder particles. These results are shown in Figure 5.

When the powder feeding rate remains constant, the focal plane of the powder flow will move from below to above the laser beam focal plane as the carrier gas flow rate increases. This is because the resistance of the carrier gas increases as the flow rate of the carrier gas increases, and the velocity of the powder particles also increases, similar to the velocity of the carrier gas. Therefore, when the powder particles left the nozzle, they had a higher degree of convergence with the cone angle, causing the focal plane to move upward. In the case of a constant carrier gas flow, the greater the powder feeding rate is, the greater the degree of divergence of the powder flow will be. At the same time, a higher powder feeding rate will significantly increase the probability of collision of the adjacent powder particles, causing them to deviate from the original trajectory in flight. Therefore, the focal motion of the powder flow was more sensitive to changes in the carrier gas flow rate than changes in the powder feed rate. In addition, the dispersion of the powder flow depended mainly on the powder feeding rate.

Figure 6 shows the coupling results of the powder particle concentration scalar of the radial plane and the laser spot determined by numerical calculation at varied carrier gas flow and powder feeding rates. Here, the radial plane is located on the plane of the laser spot (the area size is 36 mm^2^), which is indicated as the black dashed range in Figure 6.

As shown in Figure 6, when the powder feeding speed was constant, the concentration of the powder particles in the center of the spot increased with the increase in the carrier gas flow. Thus, a higher powder particle concentration was observed in the center of this laser spot as the focal plane moved upwards (Figure 5). When the carrier gas flow rate was constant, the powder concentration in the laser spot increased and then decreased with the increase in the powder feeding rate. This is because the powder feeding rate exceeds the load capacity of the carrier gas, which increases the dispersion of the powder stream and the collision probability of adjacent particles, thus reducing the powder concentration at the center of the laser spot. It is worth noting that the centrality of the powder particle concentration was the best at the carrier gas flow rate of 4.0 L/min and powder feeding rate of 2.0 g/min, while the concentration of powder particles in the center of the laser spot was the highest at the carrier gas flow rate of 4.0 L/min and the powder feeding rate of 4.0 g/min, respectively.

### 4.3. Optimal Utilization of Powder

During the LDMD process, optimal utilization can be defined as the ratio of the powder particles in the laser spot to those on the plane of the laser spot. Figure 7 shows the optimal utilization under different parameters, calculated using Adobe Photoshop CC 2019 based on the pixel value. Compared with the powder feeding rate, the carrier gas flow rate had a significant effect on optimal utilization. When the powder feeding rate was constant, the optimal utilization increased with the increase in the carrier gas flow rate. This is due to the fact that the focal plane of the powder flow moved upward so that it neighbored the laser spot (Figure 5). However, the optimal utilization first increased and then decreased with the increasing powder feeding rate, while the carrier gas flow rate was constant. As discussed in Section 4.2, the powder particles could only be effectively transported at the load capacity of the carrier gas. The optimal utilization began to decrease as it fell below the load capacity. A possible explanation for this is that part of the powder particles did not fall into the laser spot during the convergence process due to the fact that the axial velocity component was greater than the radial velocity. In addition, the calculations in Figure 7 show that the highest powder utilization of 81.4% was achieved when the powder feeding rate was 4.0 g/min and the carrier gas flow rate was 4.0 L/min.

### 4.4. Geometric Characteristics of LDMD and the Actual Utilization of Powder

The geometric characteristics of LDMD, which mainly include the width (*W*), height (*H*), and contact angle (*θ*) with the substrate, are shown in Figure 8a. The definitions of these geometric characteristics have been discussed in previous studies [36].

According to the simulation results of the spatial geometric coupling characteristics of the optical powder, the experimental conditions are as follows: keep the carrier gas flow rate constant at 4.0 L/min and change the powder feeding speed for LDMD (the black curve in Figure 8b), and keep the powder feeding speed constant at 4.0 g/min and change the carrier gas flow rate for LDMD (the red curve in Figure 8b). The geometric characteristics of LDMD and the statistical results for the actual utilization of powder particles are shown in Figure 8b. It can be seen that the *H* of the LDMD is clearly more affected by the process parameters than the *W*. In the LDMD process, the *W* is determined by the laser spot size, which fluctuates within around 0~5 mm, while the *H* is determined by the powder particle concentration in the laser spot caused by the combined effects of the carrier gas flow and powder feeding rates (Figure 6). It is worth noting that a higher actual utilization (54%) of powder could be obtained at a relatively low powder feeding rate (2.3 g/min). Then, the actual utilization began to decrease with the increase in the powder feeding rate.

## 5. Conclusions

(1) The focal plane motion of the powder stream is more sensitive to changes in the carrier gas flow rate than changes in the powder feed rate. In addition, the degree of dispersion of the powder stream is mainly determined by the powder feeding speed.

(2) When the powder feeding speed is constant, the powder concentration in the center of the laser spot increases with the increase in the carrier gas flow. Additionally, when the carrier gas flow rate is constant, the powder concentration in the center of the laser spot increases and then decreases with the increase in the feeding speed. When the carrier gas flow rate is 4.0 L/min and the powder feeding speed is 4.0 g/min, the optimal powder utilization is 81.4%.

(3) Compared with the width of the cladding layer, the effects of the carrier gas flow rate and powder feeding speed on the height of the cladding layer are more obvious. In addition, in the actual LDMD experiment, when the carrier gas flow rate was 4.0 L/min and the powder feeding speed was 2.3 g/min, the optimum powder utilization was 54%.

In this paper, we mainly studied the coupling characteristics of a powder and laser beam at different powder feeding speeds and carrier gas flow rates, focusing on the physical coupling morphology and the law of the powder and laser in space. This work lays a solid foundation for the next stage of our research, which will focus on the thermal coupling characteristics of the powder and laser.

## Figures and Tables

**Figure 1 materials-16-03403-f001:**
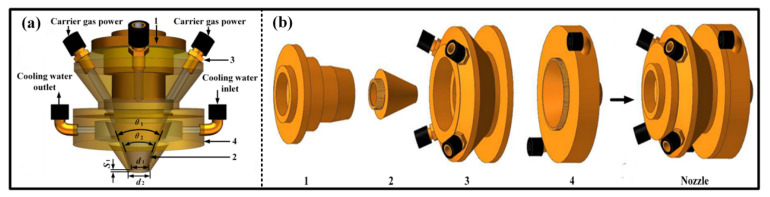
Schematic diagram of the structure of the coaxial cone nozzle. (**a**) The structure assembly diagram of the coaxial cone nozzle; (**b**) The structure explosion diagram of the coaxial cone nozzle (1: Top cover; 2 and 4: Lower cones; 3: Upper cone).

**Figure 2 materials-16-03403-f002:**
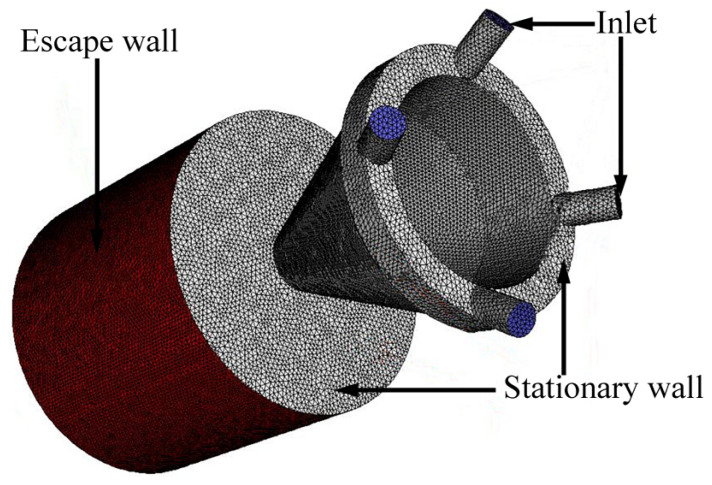
Domain mold with the meshed model of the coaxial cone nozzle.

**Figure 3 materials-16-03403-f003:**
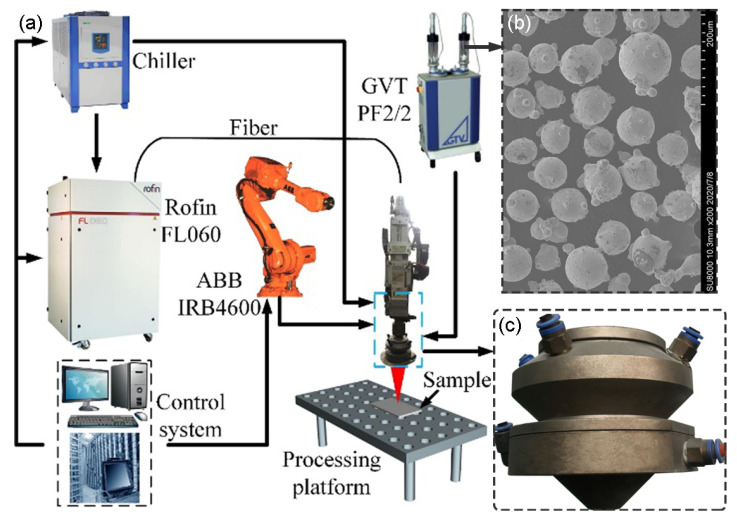
Laser processing system and powder morphology. (**a**) Rofin-FL060 6 kW; (**b**) Ni60A. (**c**) Real part of the coaxial cone nozzle.

**Figure 4 materials-16-03403-f004:**
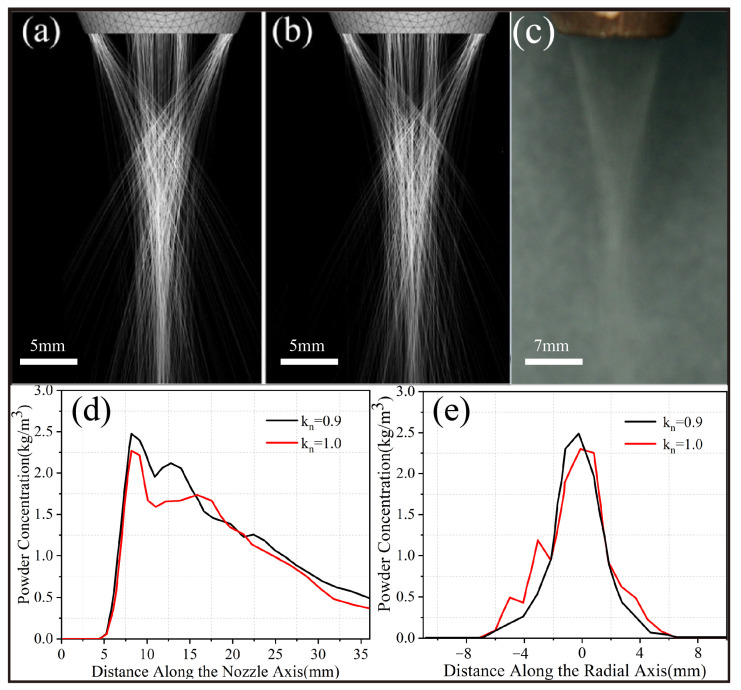
Morphology and concentration of powder flow (the carrier gas flow rate = 4.0 L/min, and the powder feeding rate = 4.0 g/min): (**a**,**b**) The morphology of powder flow at *k_n_* = 0.9 and *k_n_* = 1.0, respectively, obtained by numerical calculation; (**c**) The morphology of powder flow obtained by experimentation; (**d**,**e**) The concentration of powder at *k_n_* = 0.9 and *k_n_* = 1.0, respectively, obtained by numerical calculation.

**Figure 5 materials-16-03403-f005:**
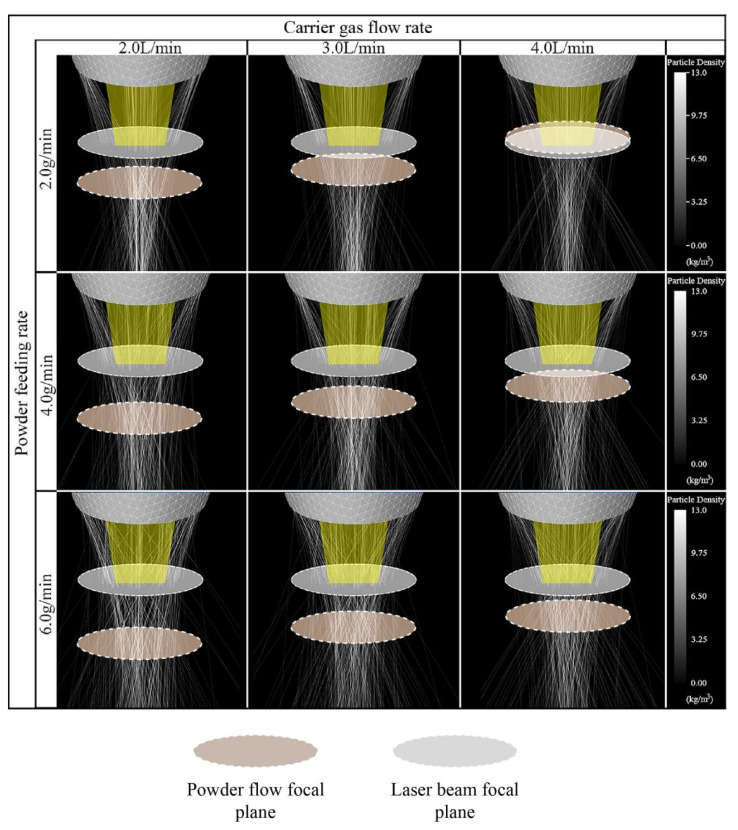
Coupling results of the axial powder particles’ trajectory and laser beam obtained by numerical calculation under different carrier gas flows (2 L/min, 3 L/min, and 4 L/min) and powder feeding rates (2 g/min, 4 g/min, and 6 g/min).

**Figure 6 materials-16-03403-f006:**
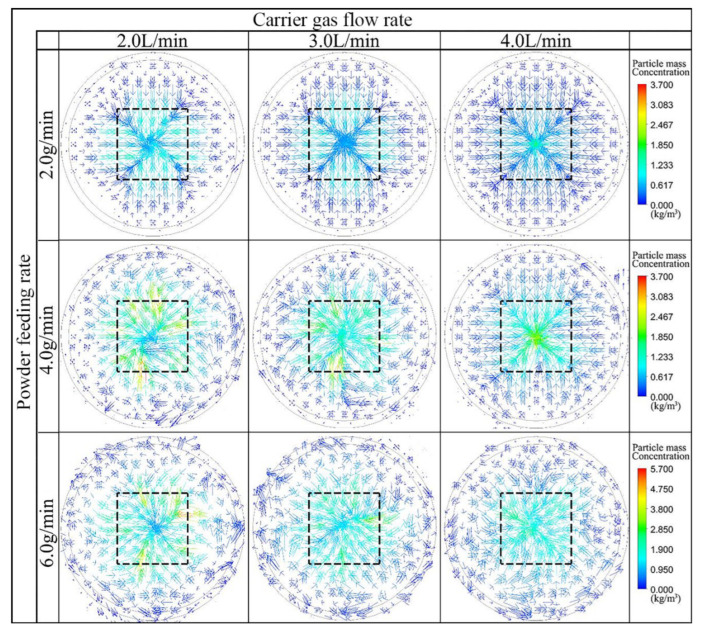
Coupling results of the powder particle concentration scalar of the radial plane and the laser spot by numerical calculation under different carrier gas flow rates (2 L/min, 3 L/min, and 4 L/min) and powder feeding rates (2 g/min, 4 g/min, and 6 g/min).

**Figure 7 materials-16-03403-f007:**
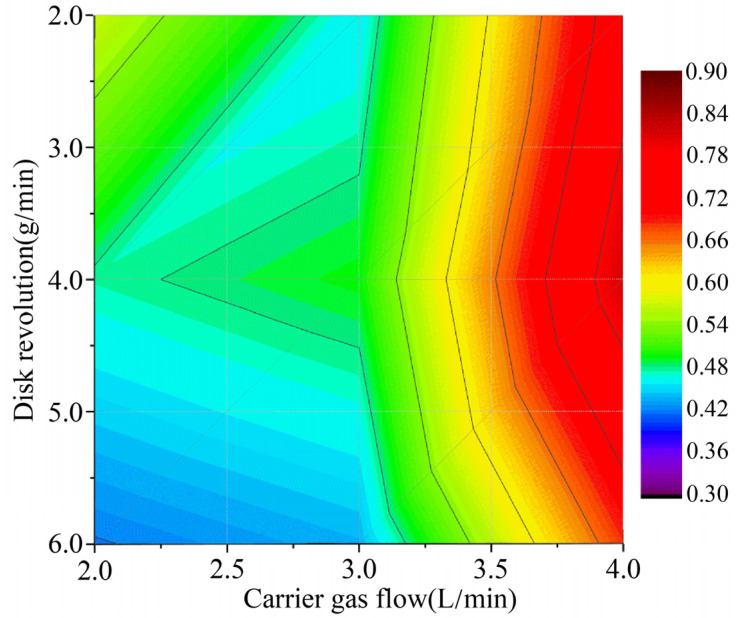
Optimal utilization under different parameters, calculated using Adobe Photoshop CC 2019 based on the pixel value.

**Figure 8 materials-16-03403-f008:**
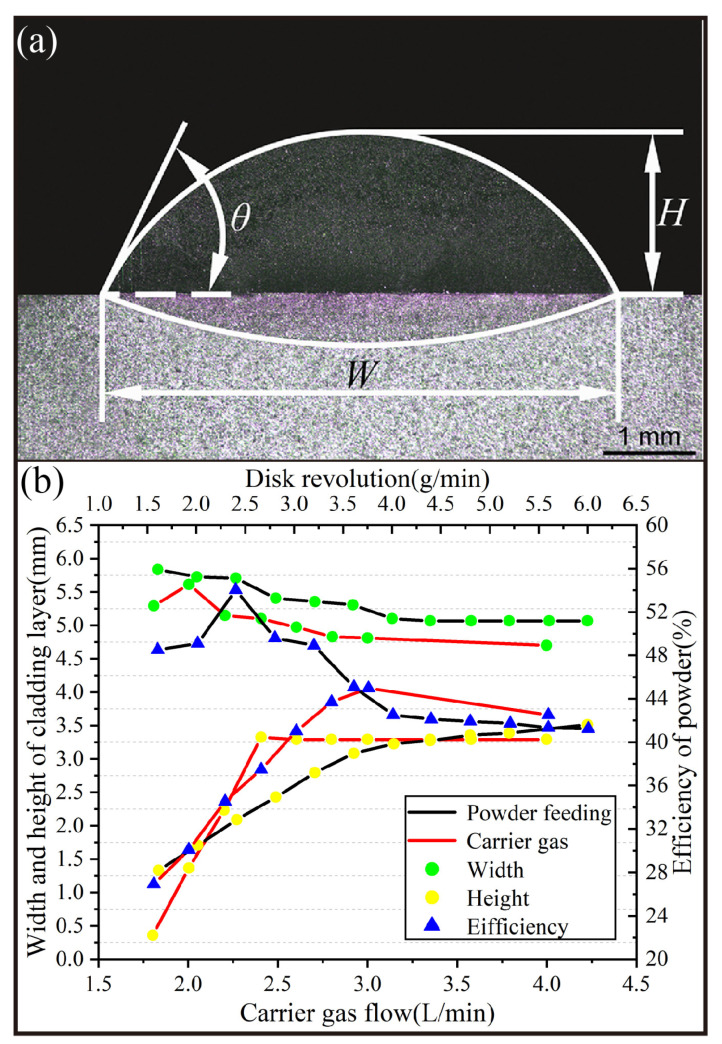
(**a**) Geometric characteristics of LDMD; (**b**) Statistical results of LDMD geometrical characteristics and actual powder utilization at different carrier gas flow rates and powder feed rates.

**Table 1 materials-16-03403-t001:** The geometric parameters of the coaxial conical nozzle.

Categories	Parameters	Values
Cones (4) angle	*θ* _1_	45°
Cones (3) angle	*θ* _2_	39.9°
Inner diameter	*d* _1_	12 mm
External diameter	*d* _2_	13.1 mm
Difference	*S* _1_	1.3 mm

**Table 2 materials-16-03403-t002:** The simulation parameters of Ni60A.

Categories	Parameters	Values
Density	*ρ_p_*	4700 kg/m^3^
Minimum diameter	*d_min_*	35 μm
Maximum diameter	*d_max_*	100 μm
Average diameter	*d_ave_*	79 μm
Gravity acceleration	*g*	−9.81 m/s^2^

**Table 3 materials-16-03403-t003:** Processing parameters of laser direct metal deposition.

Categories	Values
Laser power	3000 W
Scanning speed	6 mm/s
Defocusing amount	±1 mm
Beam size	6 × 6 mm^2^
Carrier gas flow	1.8–4.0 L/min
Powder feeding rate	1.6–6.0 g/min

## Data Availability

No new data were created or analyzed in this study. Data sharing is not applicable to this article.

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
