# Peer review of "Coupling Characteristics of Powder and Laser of Coaxial Cone Nozzle for Laser Direct Metal Deposition: Numerical Simulation and Experimental Study"

_materials, 2023, doi:10.3390/ma16093403_

Round 1

Reviewer 1 Report

The flow is represented by a steady standard k-epsilon model. An influence of the particles on the flow was not considered. In equation 6, a \partial symbol should be corrected to turbulent viscosity \mu_t. Accordingly, u is probably the mean velocity and in equation 11 the v is also u. In equation 12 a drag coefficient C_D is used which probably refers to the mean velocity. However, it is not clear how it does this. To consider a highly transient process only averaged and not even to describe the used one-way coupling I think is negligent in many respects.

Statements about the particle concentration in Fig. 4 are accordingly more than unclear, since the difference between the "concentration of powder" k_n (parameter of 4d and 4e) and the "powder concentration" as abscissa of the same graph is not sufficiently clarified. The graphs 4d, 4e and 8b are also of very poor quality.

Overall, the small number of usable statements is also to be criticized. Basically, the simulation results are summarized in Fig. 8b. Here, the carrier gas volume flow and the mass flow of the conveyed particles are compared with the cross-section of the cladding layer and the powder utilization rate. However, it is not sufficiently clarified which volume flow rates were present at the corresponding conveying rates and which particle conveying rate was present at the volume flow rates investigated. Thus, a qualitative statement of the study is not possible. However, the authors seem to be aware of this, since the summary only refers to the maximum particle utilization rate of 81%, which cannot be read at all in Fig. 8. Only in Fig. 7 can it be seen that the particle efficiency at maximum flow rate also appears to be maximum. This again makes the reader wonder if the efficiency could be increased with higher volume flows and why higher volume flows of the carrier gas were not investigated.

Reviewer 2 Report

Numbers mean lines in the article:

26: put space between W and (Width),

98: delete "." sign after ";" sign. Description of sub (b) should have the same font height as in sub "a". Here it looks like it is smaller (in pdf file),

121, 142, 145, 146, 147 etc: align formula number (1, 6, 7, 8, 9 etc) to the right,

204: Fig. 4c: It is worth put scale on this figure to compare with figures 4a and 4b.

It is good article. It is shame that Authors did not show all parameteres from numerical calculations that they used (if it is possible put the rest of parameters in the revised version). Authors should also shown all initial boundary conditions. It would be nice to know how the cooling system has influence of the results? If Authors have any information about it it is worth to write in the article. And it is worth to write what forces were not analysed and why (line: 172)?

Reviewer 3 Report

In this paper, the authors present a numeric simulation and an experimental study about coupling characteristics of powder and laser in direct metal deposition.
From my point of view there are some aspects to improve:
1. The abstract should be revised based on the following: (1) Background: Place the question addressed in a broad context and highlight the purpose of the study; (2) Methods: Describe briefly the main methods or treatments applied; (3) Results: Summarize the article's main findings; and (4) Conclusions: Indicate the main conclusions or interpretations.
2. The introduction should be improved. What are the conclusions after the current state of the research field?
3. All the methods used should be clearly explained. Thus, based on the title of the paper two main categories of methods should be explained: the methods used in the numerical simulation and the method used in the experimental study.
The authors should clearly explain the method for a good reader understanding.
4. “The coaxial cone nozzle was converted into the domain mold with the meshed in Fig.2 using the software ICEM.” Is it about of Ansys ICEM CFD? Please specify the complete name of the software. What type of mesh element was used?
5. The following paragraph should be revised. It is unclear.  What is the link between CFD simulation and “particle  size was measured..”?
“The powder particle selected for the numerical simulation is Ni60A, and its detailed simulation parameters are shown in table 2. (Table 2). The average value of the particle  size was measured with a laser granularity analyzer and this value was used to solve the  problem. First, put the dispersion medium and tested powder into the instrument, start  the ultrasonic generator to fully disperse the samples, then the circulation pump is started  and the test waits for the end. Finally, the particle size data is output using the instrument's supporting software.”
6. What is the purpose of  “2.2.Calculation parameters” section? It is unclear.
Also, please explain the results obtained by this section.
7. Regarding the section “2.3. Method of equations solution and general assumptions”.
The authors present some relations. Please keep only the new relations (proposed by the authors) or please explain the algorithm used to solve the problem.
“Equations 2-13 were solved through the built-in function of CFD software.” What kind of CFD software was used? How was implemented the “Equations 2-13” in the CFD software?
8. The Figures 4 and 8 are unclear. Please improve them.
9. The conclusions should be improved and done based o the results. Also, future research directions may also be highlighted.
10. Are the limitations of this study noted? The limitations of this study should be discussed.

Round 2

Reviewer 3 Report

The paper should be carefully improved.
All the responses of the author from the coverletter should be adapted and integrated in the paper. Thus, it is about the details from the Response 4 and Response 7.
The reference “Zekovic et al.” should be added to  Reference section.
